

# Sicegar: R package for sigmoidal and double-sigmoidal curve fitting

M. Umut Caglar, Ashley I. Teufel and Claus O. Wilke

Department of Integrative Biology, University of Texas at Austin, Austin, TX, USA

## ABSTRACT

Sigmoidal and double-sigmoidal dynamics are commonly observed in many areas of biology. Here we present sicegar, an R package for the automated fitting and classification of sigmoidal and double-sigmoidal data. The package categorizes data into one of three categories, "no signal," "sigmoidal," or "double-sigmoidal," by rigorously fitting a series of mathematical models to the data. The data is labeled as "ambiguous" if neither the sigmoidal nor double-sigmoidal model fit the data well. In addition to performing the classification, the package also reports a wealth of metrics as well as biologically meaningful parameters describing the sigmoidal or double-sigmoidal curves. In extensive simulations, we find that the package performs well, can recover the original dynamics even under fairly high noise levels, and will typically classify curves as "ambiguous" rather than misclassifying them. The package is available on CRAN and comes with extensive documentation and usage examples.

## INTRODUCTION

Growth patterns resembling sigmoidal or double-sigmoidal curves are common in biological systems (*DeLean, Munson & Rodbard, 1978*). In population biology, the number of individuals in a population frequently follows a sigmoidal growth pattern (*Lefkovitch, 1965*; *Tsoularis & Wallace, 2002*), and in molecular systems the production of proteins has similar dynamics (*Tholudur, Ramirez & McMillan, 1999*; *Doepel et al., 2004*). In virology, infections with fluorescently labeled viruses cause an initial sigmoidal growth as viral protein is produced and subsequently a sigmoidal decay as the infected cells lyse and the fluorescent protein disperses into the media (*Guo et al., 2017*). Quantifying and comparing these kinds of growth dynamics can lead to valuable insight into the systems' behavior. However, the automated quantification and classification of thousands of noisy growth curves, in a high-throughput manner, can be challenging, despite extensive literature on fitting s-shaped data (*Motulsky & Christopoulos, 2004*; *Ritz & Spiess, 2008*; *Meddings, Scott & Fick, 1989*). The challenge is that non-linear curve-fitting algorithms generally only guarantee convergence to a local minimum, and therefore in high-throughput settings there will often be some input datasets for which the fitting algorithm produces undesired results or doesn't converge.

There are several existing software packages that provide sigmoidal and double-sigmoidal curve fitting. Within the R software ecosystem, the package `drc` can fit sigmoidal and biphasic curves, and it is widely used for analysis for many different types of

Corresponding author
M. Umut Caglar,
umut.caglar@gmail.com

data (*Ritz et al., 2015*). Several other R packages are focused on dose–response optimization and curve fitting, including qpcR (*Ritz & Spiess, 2008*), grofit (*Kahm et al., 2010*), FlexParamCurve (*Oswald et al., 2012*), drfit (*Ranke, 2006*), and MCPMod (*Bornkamp, Pinheiro & Bretz, 2009*). Here, we address two problems in sigmoidal curve fitting that aren't fully covered by these packages: (i) reliable, automated fitting of thousands of sigmoidal and double-sigmoidal curves with minimal human supervision; and (ii) automated classification of measured time courses into either sigmoidal or double-sigmoidal patterns, as well as automated classification of time courses that cannot reliably be fit by a sigmoidal or double-sigmoidal model.

Our approach is implemented in the R package sicegar. The input data is assumed to represent intensity measured over time, where intensity represents any metric of a system that may vary in a sigmoidal or double-sigmoidal pattern. Using these time course data, sicegar categorizes the data as "sigmoidal," "double-sigmoidal," "ambiguous," or "no signal," by first checking the data for the presence of signal (i.e., the intensity curve deviates significantly from a horizontal line and varies significantly) and then fitting two mathematical models to the data, a sigmoidal one and a double-sigmoidal one. The package automatically determines which curve provides the better fit, and it reports a wide range of parameters about the fitted models that enable easy comparison of the fitted curves within and among experiments.

The sicegar package was originally written to study poliovirus infection and replication at the single-cell level (*Guo et al., 2017*), and the package name is inspired by this application (SIngle CEll Growth Analysis in R). In *Guo et al. (2017)*, we measured poliovirus replication in high-throughput using a microfluidics device that allowed isolation and monitoring of individual, infected cells. Poliovirus was tagged with green-fluorescent protein, and fluorescence microscopy then enabled us to observe the progress of the poliovirus infections in the cells. As the virus began replicating inside a cell, we generally saw an initial exponential increase in fluorescence that eventually leveled off. If a cell lysed within the duration of the experiment, then we saw subsequent decrease in fluorescence, resulting in an overall double-sigmoidal curve, but if the cell didn't lyse then the fluorescence time course was sigmoidal. For cells that did not experience a successful infection, time courses displayed neither a sigmoidal nor a double-sigmoidal pattern. However, measured fluorescence intensities were never exactly zero, due to random fluctuations in background fluorescence. Therefore, it was critical that our algorithm could reliably tell sigmoidal and double-sigmoidal patterns from other patterns.

While sicegar was developed in the context of single-cell viral infections, its potential applications go beyond this one use case, and the package is designed as a generalized way to classify growth data from time course experiments. In the following, we describe the mathematical models used to classify the data, how the models are fit, and the inferences that can be made from them.

## METHODS

Growth in many biologically systems can be though of as occurring in two phases. In the first phase, intensity will increase exponentially until saturation is reached and it levels off

at some asymptotic maximum level. In the second phase, the intensity may then decay from the maximum value to a lower one or even back to zero. Depending on the system and/or the length of the observation this second phase may or may not occur. These two phases can be modeled with two sigmoidal curves that describe the relationship between time and intensity. The first describes the increase of intensity and the second its decay.

Based on this two-phase view of growth an observer of a system may observe three different types of behaviors of intensity over time. First, the observer may observe the full process and clearly see both the growth and the decay phases, i.e., the full double-sigmoidal curve. Second, if the observation ends before decay occurs, or if the decay phase is not relevant for the system under study, the observer may only see the first sigmoidal curve. Third, if the intensity is not produced in detectable amounts, the observer will not see any signal at all. We label these three types of behavior as "double-sigmoidal," "sigmoidal," and "no signal." If a given time course does not clearly fit into any of these three categories we label it as "ambiguous."

## Sigmoidal and double-sigmoidal models

The simplest sigmoidal curve can be uniquely determined by three parameters, the maximum value, the midpoint, and the slope of the curve (Fig. 1A). To represent such a sigmoidal curve, the `sicegar` package uses the logistic function (*Verhulst, 1845*)

$$I(t) = f_{\text{sig}}(t) = \frac{I_{\max}}{1 + \exp(-a_1(t - t_{\text{mid}}))}. \tag{1}$$

Here, $I(t)$ is the intensity time course, given as a function of time $t$. The three parameters to be fitted are $I_{\max}$, $t_{\text{mid}}$, and $a_1$. The parameter $I_{\max}$ represents the maximum intensity observed, the parameter $t_{\text{mid}}$ indicates the time at which intensity has reached half of its maximum, and the parameter $a_1$ is related to the slope of $I(t)$ at $t = t_{\text{mid}}$ via the formula $d/dt\, I(t)t = t_{\text{mid}} = a_1 I_{\max}/4$ (Fig. 1A).

A double-sigmoidal curve will generally need at least six parameters, two midpoints and slopes parameters each, a maximum value, and an asymptotic final value after decay (Fig. 1B). We define the double-sigmoidal model by combining two regular sigmoidal functions. This combined function rises from 0 to a maximum value $I_{\max}$ at time $t^*$ and then decays to the final value $I_{\text{final}}$. The function is divided into two parts, one to the left of the maximum, for $t < t^*$ (the growth phase) and one to the right of the maximum, for $t > t^*$ (the decay phase). The two parts are rescaled separately, such that the maximum value of $I(t)$ is given by $I_{\max}$ and the final value of $I(t)$ for large times is given by $I_{\text{final}} > 0$.

To define our double-sigmoidal model, we begin with a simple double-sigmoidal function, obtained by multiplying together two sigmoidal functions:

$$f_{\text{dsig-base}}(t) = \frac{1}{1 + \exp(-a'_1(t - t'_{\text{mid1}}))} \frac{1}{1 + \exp(-a'_2(t - t'_{\text{mid2}}))}. \tag{2}$$

We then define the time $t^*$ at which this function is maximal,

$$t^* = \operatorname{argmax} f_{\text{dsig-base}}(t), \tag{3}$$
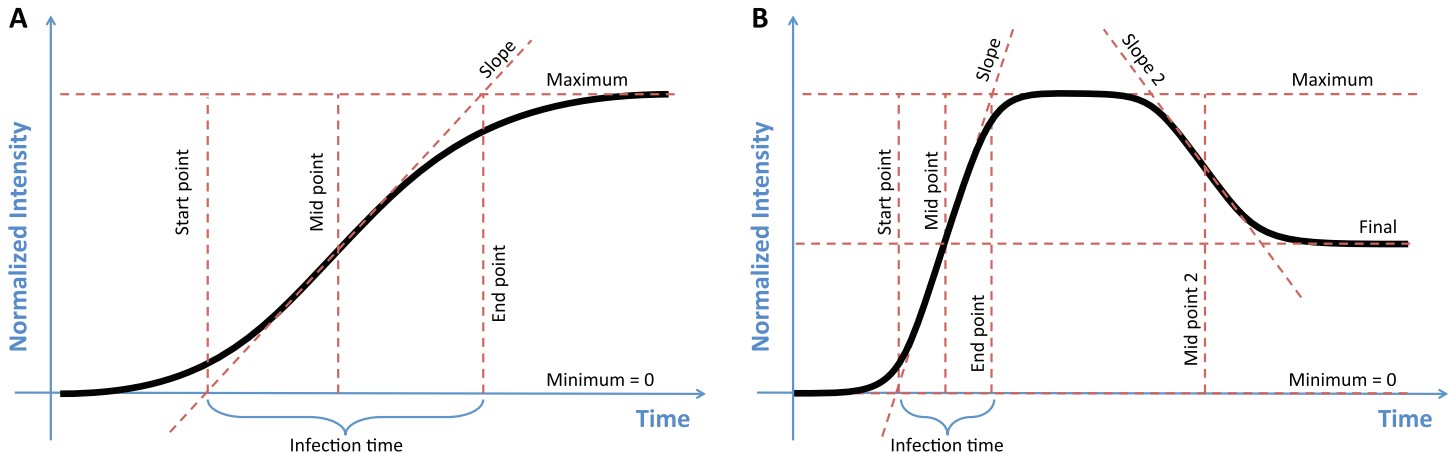

**Figure 1 The two models used to classify growth data, with key parameters of each model labeled.** (A) In the sigmoidal model the maximum value, the midpoint, and the slope of the curve uniquely describe the function. Estimates of the start and end points of growth, as well as the growth time, can then be calculated from this model. (B) The double-sigmoidal model is uniquely determined by six parameters, two midpoint and two slope parameters, a maximum value, and a final value. As in the case of the sigmoidal model, the start point, end point, and duration of both the growth and the decay phases can be calculated from these parameters.

and the value of the function at that time point,

$$f_{\text{max}} = f_{\text{dsig-base}}(t^*). \tag{4}$$

We then write

$$I(t) = f_{\text{dsig}}(t) = \begin{cases} c_1 f_{\text{dsig-base}}(t) & \text{for } t \leq t^* \text{ (growth phase)} \\ c_2 f_{\text{dsig-base}}(t) + I_{\text{final}} & \text{for } t > t^* \text{ (decay phase)} \end{cases}, \tag{5}$$

with

$$c_1 = \frac{I_{\text{max}}}{f_{\text{max}}} \tag{6}$$

and

$$c_2 = \frac{(I_{\text{max}} - I_{\text{final}})}{f_{\text{max}}}. \tag{7}$$

This definition may seem awkward and overly complex, but it guarantees that certain non-sigmoidal corner cases are eliminated (see Appendix).

The meaning of the parameters in the double-sigmoidal model mirror those of the regular sigmoidal model. The time point $t'_{\text{mid1}}$ determines (but does not exactly represent) the time at which the intensity has risen to half of its maximum, and the time point $t'_{\text{mid2}}$ (enforced to be larger than $t'_{\text{mid1}}$) determines (but does not exactly represent) the time at which the intensity has decayed halfway from its maximum to its final value. The prime in $t'_{\text{mid1}}$ and $t'_{\text{mid2}}$ indicates that these time points are not the final parameters of interest. The true midpoints $t_{\text{mid1}}$ and $t_{\text{mid2}}$ are calculated numerically by `sicegar` and reported. Similarly, the parameters $a'_1$ and $a'_2$ determine the speed of growth and speed of decay, respectively, but do not represent the slopes at times $t_{\text{mid1}}$ and $t_{\text{mid2}}$. The true midpoint slopes are calculated by numerically

differentiating $f_{dsig}$ at each midpoint time. As stated before, the parameters $I_{max}$ and $I_{final}$ represent the maximum intensity and the final intensity after decay, respectively.

In addition to the parameters that uniquely determine the sigmoidal and double-sigmoidal functions, the `sicegar` package calculates other useful metrics that describe aspects of the curves (Fig. 1). For the sigmoidal model, we calculate the starting time of growth by finding the point that a line which passes though the midpoint $t_{mid}$ with the slope $a_1$ intersects with the $x$-axis. This value gives the length of time between the start of observations and the first observable increase in the response variable. Using the same line which passes through $t_{mid}$ with slope $a_1$, we can also determine the amount of time that has elapsed during exponential growth, by looking at the time point where the line intersects with the maximum intensity ($I_{max}$) and subtracting the starting time. For the double-sigmoidal function the time until decay can be calculated in a similar way (Fig. 1B). We chose to calculate these specific start and end time points for their simple geometric interpretation. They tend to provide a conservative estimate for the total duration of growth or decay.

## Model fitting and categorization

We implemented R code that can fit data to the sigmoidal and double-sigmoidal curves defined by Eqs. (1) and (5). Before fitting, data is internally normalized to a [0,1] interval on both the $x$- and $y$-axis. The sigmoidal and double-sigmoidal models are then fit to the internally normalized data via likelihood maximization, using the function `nls.lm` in the `minpack.lm` package (*Elzhov et al., 2015*). The `minpack.lm` package uses a modified Levenberg–Marquardt algorithm (*Moré, 1978*) to minimize the sum of the squared residuals. To guarantee robust fitting, we run `nls.lm` multiple times using different starting estimates, chosen at random from a uniform distribution covering the allowed interval of parameters. For each fit, the quality of fit based on Akaike information criterion (AIC) (*Akaike, 2011*; *Burnham & Anderson, 2004*) scores is recorded, and the parameter estimates that produce the overall best fit across replicates are reported as the final estimates.

The simplest way to use `sicegar` is via the function `fitAndCategorize`, which attempts to both identify the best-fitting model and determine whether this model is sigmoidal, double-sigmoidal, or neither. This function first checks whether there is any signal in the data, by testing whether the maximum observed intensity exceeds a user-defined threshold and the data range also exceeds a user-defined threshold. If either test fails, the data are labeled as "no signal."

Next, the function fits both a sigmoidal and a double-sigmoidal curve to the data and then assess which provides the better fit. This process considers a number of different factors. First, it assesses whether both models could actually be fitted (i.e., the fitting did not fail). Second, the AIC scores of the fitted models are examined to make sure they are smaller than a threshold value (the default is −10). Third, the starting time points of each model are checked to ensure that they are positive and the starting intensities are examined to ensure that they are smaller than a threshold (the default is 0.05). Fourth, the ratio of the models' intensity prediction at the last observation to the

models' maximum intensity prediction is tested to ensure that it is over 0.85 in the case of the sigmoidal model and below 0.75 in the case of the double-sigmoidal model. These various checks and cutoffs are heuristics which we developed by manually inspecting thousands of fits in the context of our single-cell virology study (*Guo et al., 2017*). Some of these (in particular the AIC cutoff) may not be appropriate for a very different dataset and may require adjustment. Therefore, none of these cutoffs are hardcoded into `sicegar` and they can all be adjusted as needed.

If any of the above tests fail for either the sigmoidal or double-sigmoidal fit, then the data is not categorized as being described by that model. If both models fail any of these tests the data is categorized as being "ambiguous." If only one model passes these tests while the other fails, then the data is considered to be best described by that model. If both of the fits pass all of these test a decision is made about which model best describes the data based on their AIC scores. If the sigmoidal AIC score is smaller than the double-sigmoidal AIC score then the data is labeled as "sigmoidal," if it is larger then the data is labeled as "double-sigmoidal." There is also an option for favoring one or the other model by adding a bonus value to the AIC score of either the sigmoidal or the double-sigmoidal model. By default, the bonus is turned off (set to zero).

Users can modify the complete decision process, as well as the specific parameters to adjust the package to their own needs, as explained in detail in the vignette "Identifying the best-fitting model category" supplied with the R package.

## Simulations

To assess `sicegar` performance, we simulated sigmoidal and double-sigmoidal data with different levels of noise. We first generated sets of parameters for both sigmoidal and double-sigmoidal curves. For the sigmoidal model, the maximum and midpoint parameters were chosen uniformly at random from the intervals $[0.3, 20]$ and $[3, 27]$, respectively. Slopes were generated by first drawing a random slope angle $\theta$ uniformly from the interval $[0, \pi/2]$ and then converting into a slope via $\tan(\theta)$. For the double-sigmoidal model, the maximum, the first midpoint, the decreasing slope, the distance between midpoints, and the final asymptote intensity ratio parameters were chosen uniformly at random from the intervals $[0.3, 20]$, $[3, 26]$, $[0.001, 40]$, $[1, 27—first\ midpoint]$, and $[0, 0.85]$, respectively. The increasing slope was generated as in the sigmoidal model, by uniformly sampling the $\theta$ interval $[0, \pi/2]$. We excluded all randomly generated parameter sets that by definition would be classified as ambiguous by `sicegar`, for example because the difference between the maximum and minimum intensity fell below our pre-defined cutoff.

For each parameter set, we simulated data for a time interval of 27 units, from $t = 3$ to $t = 30$. We used five different temporal sampling approaches, and in each case generated time points and then the associated intensity values according to the chosen model and model parameters. The sampling approaches we considered were: (i) equidistant sampling, where we generated 55 time points in increments of 0.5; (ii) random sampling from a uniform distribution, where we generated 55 time points uniformly distributed from 3 to 30; (iii)–(v) random sampling from beta distributions, creating higher sampling

densities at the beginning, center, and end of the time course. Here, we set the $\alpha$ and $\beta$ parameters of the beta distributions to {$\alpha = 0.5$, $\beta = 1.5$} (mode at the beginning), {$\alpha = 2$, $\beta = 2$} (mode at the center), and {$\alpha = 1.5$, $\beta = 0.5$} (mode at the end), respectively, and sampled 55 time points from the respective beta distribution and then multiplied them by 27 and added 3, so all time points fell into the interval from 3 to 30.

For each generated intensity curve, we then created noisy variants. We used 11 distinct noise amplitudes, ranging from 0% to 150% of the maximum value of the initial simulated data. To generate additive noise, we created vectors of 55 random numbers chosen uniformly from −0.5 to 0.5. We then multiplied each random number with the noise amplitude (number between 0 and 1.5) and the maximum intensity of the dataset. Finally, we added these noise values to the intensity values of the initial samples and obtained noisy samples. To generate multiplicative noise, we generated vectors of 55 random numbers $2^x$, where $x$ was uniformly distributed between −1 and 1. We then multiplied each random number with the noise amplitude, and we then multiplied the noise vector element-wise with the original intensity vector.

We generated 50 replicate parameter sets for each combination of model (sigmoidal or double-sigmoidal), temporal sampling approach, noise amplitude, and noise type (additive or multiplicative). We generated a new set of random parameters in each case, so that in total we evaluated $50 \times 2 \times 5 \times 11 \times 2 = 11,000$ different parameter sets.

For each simulated sample, we used `sicegar` to estimate both the shape of the curve ("sigmoidal," "double-sigmoidal," or "ambiguous") and the curve's original parameter values. To estimate the algorithm's accuracy in fitting a model to the data, we calculated the mean absolute error between the initial (without noise) and predicted intensity vectors, and we normalize this number relative to the maximum intensity of the initial sample. We calculated the normalized mean absolute error regardless of whether the algorithm predicted the category of the sample correctly or not.

### Package availability and documentation

The `sicegar` package is implemented in R and available from CRAN at https://CRAN.R-project.org/package=sicegar. The `sicegar` source code is available on github at https://github.com/wilkelab/sicegar. Extensive user documentation is available in multiple vignettes distributed as part of the package.

The package depends on existing R packages `dplyr` (*Wickham & Francois, 2015*), `minpack.lm` (*Elzhov et al., 2015*), `fBasics` (*Wuertz, Setz & Chalabi, 2014*), and `ggplot2` (*Wickham, 2009*).

## RESULTS AND DISCUSSION

### Performance against simulated data

Results from our simulation benchmark are shown in Fig. 2 (for equidistant temporal sampling and additive noise only) and in Fig. S1 (for all five sampling schemes and both additive and multiplicative noise). We first asked how well `sicegar` recovers the type of the curve (sigmoidal or double-sigmoidal) at varying noise levels (Fig. 2A; Fig. S1A). For equidistant sampling with additive noise, we found that curves that were

simulated with sigmoidal input were virtually never classified as double-sigmoidal (Fig. 2A). However, starting at noise levels of around 45% (corresponding to a noise amplitude of half the total amplitude of the sigmoidal curve), `sicegar` classified a fraction of the curves as ambiguous. The fraction of curves classified as ambiguous increased with increasing noise level until virtually all curves were classified as ambiguous at noise levels of around 100%. Results were similar for curves simulated as double-sigmoidal, except that a small fraction of curves was always identified as sigmoidal, regardless of the noise level (Fig. 2A). These were curves for which the parameter values happened to cause very little decay at large times, either because the final asymptotic value was close to the maximum or because decay started late relative to the maximum time point simulated. Finally, at high noise levels, double-sigmoidal curves can appear sigmoidal, and `sicegar` classified those curves as such. In general, decreased performance with increased noise level is an expected behavior (*Spiess & Neumeyer, 2010*), but our algorithm's tendency to label misclassified cases as ambiguous instead of making wrong decisions shows the robustness of the package's decision process.

Results were broadly similar for other temporal sampling regimes and multiplicative instead of additive noise (Fig. S1A), with two caveats. First, `sicegar` was generally more robust to multiplicative noise than to additive noise, though some sigmoidal curves were classified as double-sigmoidal rather than ambiguous under multiplicative noise. Second, for non-equidistant temporal sampling, the performance was best when sampling was uniform or biased toward the middle of the time interval, rather than toward the beginning or end of the interval.

We also compared the fits `sicegar` obtained from noisy data to the initial curves before applying noise. For each fit, we calculated the normalized mean absolute error, which is the mean of the absolute differences of fitted and original intensity levels, normalized by the maximum original intensity for all five different sampling regimes and for two different noise regimes. We found that these errors were near zero for low noise levels and increased gradually as noise increased (Fig. 2B; Fig. S1B). This was true for both additive and multiplicative noise and for all five temporal sampling regimes. Overall, we can conclude that our algorithm results in reliable fits, that it fails gradually with increasing noise levels, and that it is conservative in assessing whether it has correctly identified a sigmoidal or double-sigmoidal curve or not.

## Comparison to existing packages

On the face of it, fitting a sigmoidal curve to time-course data appears to be a simple and solved problem. The simplest sigmoidal curve is determined by only three parameters, and many non-linear curve-fitting algorithms exist that can be used to fit a simple three-parameter function. However, when we attempted to fit thousands of curves to experimentally measured data, we found that achieving numerical reliability was non-trivial. While a simple algorithm will work most of the time, it will fail eventually on cases that by manual inspection have a clear solution. The problem was further exacerbated for double-sigmoidal curves, where even guaranteeing that the curve is double-sigmoidal at all times was non-trivial (see Appendix). Finally, for our original

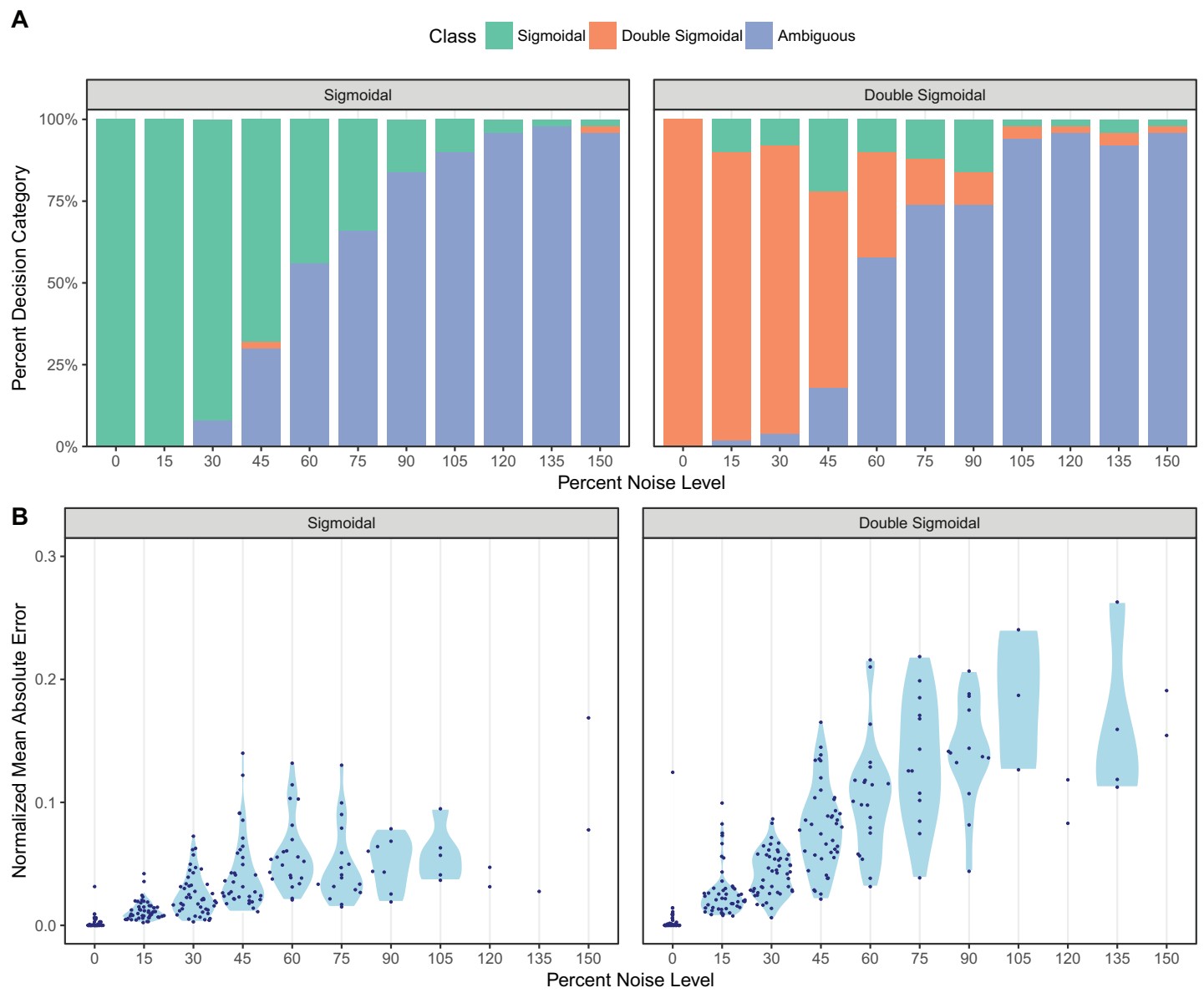

**Figure 2 Assessment of `sicegar` performance on simulated data subjected to varying levels of noise.** Noise levels are measured relative to the maximum intensity of the simulated curves. (A) `sicegar` tends to either recover the original type of curve correctly (for small to moderate amounts of noise) or label the dataset as ambiguous (at high noise levels). (B) The difference between the original data (without noise) and the fitted curve is near zero for low noise levels and increases gradually and slowly for higher noise levels.

application of single-cell poliovirus infections (*Guo et al., 2017*), we wanted to classify sigmoidal and double-sigmoidal curves in an automated fashion, and this problem goes beyond just fitting the curves to requiring a heuristic that can automatically classify curves and also determine when the classification is ambiguous.

We wrote the `sicegar` package to solve these challenges. However, `sicegar` is not the only package with the ability to fit data to such non-linear models. The qpcR (*Ritz & Spiess, 2008*) package fits linear, exponential, a number of sigmoidal, and log–logistic

functions to qPCR (realtime quantitative polymerase chain reaction) data. Unlike `sicegar` this package uses a log–logistic function rather than a double-sigmoidal function to describe data that includes both a growth and a decay phase. A similar package, `grofit` (*Kahm et al., 2010*), uses a number of parametric growth curves and a model-free spline approach to estimate parameters from dose to response data. This package does not include a parametric model that would account for a decay phase in the data, though an interface to construct custom models is provided. The intended use of this package is to model pharma-/toxicological data and it produces estimates of values such as half maximum effective concentration (EC50). While `sicegar` can estimate the midpoint, which corresponds to the EC50, `sicegar` cannot currently estimate time points corresponding to other specific intensity values along the curve. Another package, `FlexParamCurve` (*Oswald et al., 2012*), fits in excess of 30 models to data. While this package is useful for teasing apart exactly which variant of a model best describe dynamics, `sicegar` is better suited for classifying data as either containing sigmoidal or double-sigmoidal dynamics.

We can think of several future extensions to the `sicegar` package. First, instead of the three-parameter logistic function, we could describe the sigmoidal model via the five-parameter Richards' equation (*Richards, 1959*), which is currently available in the `growthmodels` package. For completeness, however, we would then also have to develop a double-sigmoidal version of this model. Second, future work could add the functionality required to calculate the time points corresponding to arbitrary percentages of the maximum effective concentration, such as the 5th percent (EC5) and the 95th percent (EC95), which could be used as alternative to the start and end point we currently calculate.

# APPENDIX: PROPERTIES AND DERIVATIONS OF THE SIGMOIDAL AND DOUBLE-SIGMOIDAL MODELS

The `sicegar` package uses two distinct models that represent growth and growth-like data obtained from time course experiments. The sigmoidal model describes situations in which the signal starts from zero and rises to a maximum level, and the double-sigmoidal model describes situations in which signal starts from zero, rises to a maximum level, and then declines toward an asymptotic value below the maximum.

Two factors were considered when choosing the corresponding mathematical functions. First, the models should be as simple as possible while being well-behaved. Well-behaved implies that the models should not take on unexpected shapes under specific parameter combinations, they should have few variables, they should be continuous, and they should be defined for all time points from minus infinity to plus infinity. As we will see below, achieving these constraints is non-trivial for the double-sigmoidal model. Second, as much as possible, model parameters should have simple interpretations and so represent or be related to biologically meaningful parameters.

## Properties of the sigmoidal function

The `sicegar` package uses for its sigmoidal fit the Fermi function (Eq. 1). Here we describe the properties of a more general variant of this function,

$$I(t) = \frac{I_{max} - I_{init}}{1 + \exp[-a_1(t - t_{mid})]} + I_{init}. \tag{8}$$

The Fermi function is recovered by setting $I_{init} = 0$.

The more general function discussed here has four parameters, each of which has a clear meaning in terms of the shape of the function:

1. $I_{init}$ represents the initial value of the sigmoidal curve at minus infinity,

$$I_{init} = \lim_{t \to -\infty} I(t). \tag{9}$$

2. $I_{max}$ represents the maximum value of the curve at plus infinity,

$$I_{max} = \lim_{t \to \infty} I(t). \tag{10}$$

3. $t_{mid}$ represents the time point at which the slope of $I(t)$ is maximal,

$$t_{mid} = \text{argmax} \frac{d}{dt} I(t). \tag{11}$$

4. The parameter $a_1$, constrained to be positive at all times, is related to the maximum slope of $I(t)$,

$$\frac{a_1}{4}(I_{max} - I_{init}) = \frac{d}{dt} I(t)\big|_{t=t_{mid}} \tag{12}$$

5. In the following, we will also need the inverse sigmoidal function,

$$I(t) = \frac{I_{max} - I_{final}}{1 + \exp[a_2(t - t_{mid})]} + I_{final}, \tag{13}$$

which differs from Eq. (8) in that it does not have a minus sign in front of the slope parameter $a_2$. We also have renamed $I_{init}$ as $I_{final}$, to indicate that it now corresponds to the limit $t \to \infty$. The inverse sigmoidal function represents sigmoidal decay rather than sigmoidal growth, starting at its maximum and decaying to its final value.

## Properties of the double-sigmoidal function

The double-sigmoidal function is defined in Eq. (5) in the main text. It rises from zero to a maximum value and then decays toward an asymptotic final value, and it is defined for all $t$. The function is continuous with a continuous first derivative and a discontinuous second derivative.

The parameters $I_{max}$ and $I_{final}$ represent the maximum and asymptotic values of the curve, respectively. However, the parameters $t'_{mid1}$, $t'_{mid2}$, $a'_1$, and $a'_2$ do not directly represent midpoints and slopes. Instead, the two midpoints and two slopes need to be calculated numerically.

## Derivation of the double-sigmoidal function

The double-sigmoidal as defined in Eq. (5) may seem overly complex and awkward. However, we had to define it in this way to avoid problems that arise with other formulations.

In particular, a seemingly more straightforward double-sigmoidal could be obtained by multiplying together a regular sigmoidal (Eq. 8) and an inverse sigmoidal (Eq. 13):

$$I(t) = \left( \frac{I_{\max} - I_{\text{init}}}{e^{-a_1(t-t_{\text{mid1}})} + 1} + I_{\text{init}} \right) \left( \frac{1 - I_{\text{final}}}{e^{a_2(t-t_{\text{mid2}})} + 1} + I_{\text{final}} \right) \qquad (14)$$

Even though this function behaves appropriately for most parameter choices, there are parameter regions in which this function is not a double-sigmoidal. This occurs in general when the interval between $t_{\text{mid1}}$ and $t_{\text{mid2}}$ is very small, and one of the slopes is very steep while the other one is not. As an example, consider the case shown in Fig. 3, where Eq. (14) generates a local minimum in addition to a local maximum. These kinds of corner cases will be discovered by the fitting algorithm when used on sufficiently many diverse datasets, and therefore we ruled out Eq. (14) as an appropriate choice for our double-sigmoidal function.

However, Eq. (14) will never have a local minimum if we set both $I_{\text{init}}$ and $I_{\text{final}}$ to zero (see next section for proof). Thus, we define the *base double-sigmoidal* as

$$f_{\text{dsig-base}}(t) = \frac{1}{(e^{-a_1(t-t_{\text{mid1}})} + 1)(e^{a_2(t-t_{\text{mid2}})} + 1)} \qquad (15)$$

This function has the correct shape in principle, except that the intensity always decays back to zero for large $t$.

One possible solution to the problem that the function approaches zero for $t \to \infty$ is to cut it into two parts, rescale them separately, and then merge them together again. We cut the function at its maximum, where $t = t^*$. Thus, the part from $t = -\infty$ to $t = t^*$ represents sigmoidal growth and the part from $t = t^*$ to $t = \infty$ represents sigmoidal decay. Both parts are rescaled such that the desired $I_{\max}$ and $I_{\text{final}}$ are obtained. This procedure leads to the definition of the double-sigmoidal we use, as defined in Eq. (5). This function has the desired limiting properties and is guaranteed to have only one maximum. However, the cutting and rescaling procedure causes its second derivative to be discontinuous at $t = t^*$.

## The base double-sigmoidal has exactly one local maximum

For the base double-sigmoidal as defined in Eq. (15) we can prove the following Lemma.
**Lemma** *$f_{\text{dsig-base}}(t)$ has one local maximum.*
*Proof.* At its local maximums and minimums, i.e., when $t = t^*$, the derivative of $f_{\text{dsig-base}}(t)$ must be equal to zero,

$$\frac{d}{dt} f_{\text{dsig-base}}(t)\big|_{t=t^*} = 0. \qquad (16)$$

By definition, $t_{\text{mid2}} > t_{\text{mid1}}$, so without loss of generality we can write $t_{\text{mid2}} = t_{\text{mid1}} + L$, where $L$ is a positive number. Then we obtain

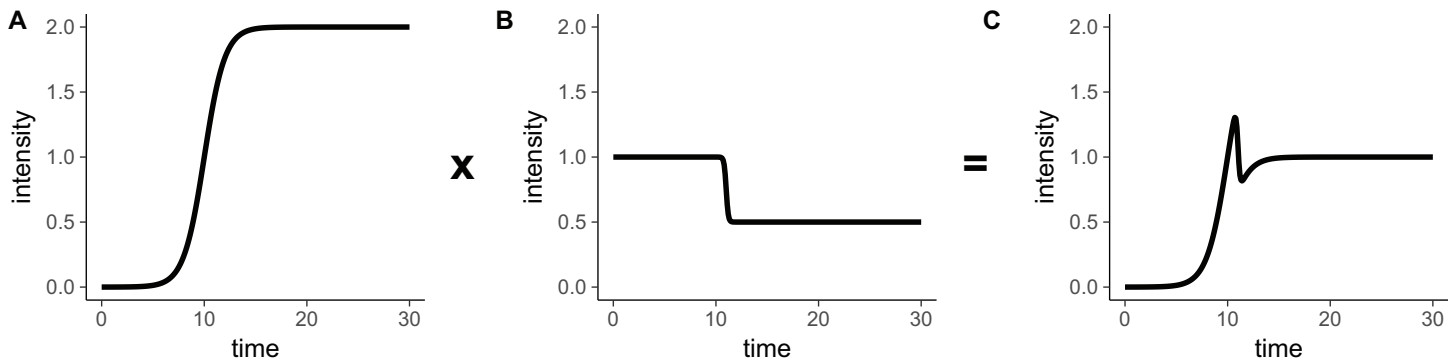

**Figure 3 The product of a rising sigmoidal and a decaying sigmoidal (Eq. 14), is not guaranteed to be double-sigmoidal.** For some parameter settings, when we multiply (A) the sigmoidal function (Eq. 8) and (B) the inverse sigmoidal (Eq. 13) function, we obtain (C) a result that is not double-sigmoidal. Parameter choices for this example are: $I_{\text{init}} = 0$, $I_{\text{final}} = 0.5$, $I_{\text{max}} = 2$, $a_1 = 1$, $a_2 = 10$, $t_{\text{mid1}} = 10$, $t_{\text{mid2}} = 11$.

$$\frac{d}{dt}\left[\frac{1}{(e^{-a_1(t-t_{\text{mid1}})} + 1)(e^{a_2(t-L-t_{\text{mid1}})} + 1)}\right]\bigg|_{t=t^*} = 0 \tag{17}$$

We define new variables $u \equiv t - t_{\text{mid1}}$ and $u^* \equiv t^* - t_{\text{mid1}}$. We also define the function $g(u) = f_{\text{dsig-base}}(u + t_{\text{mid1}})$. Finding the roots of $g(u)$ is equivalent to finding the roots of $f_{\text{dsig-base}}(t)$, and it involves solving the following equation for $u^*$:

$$\frac{d}{du}g(u)\bigg|_{u=u^*} = \frac{d}{du}\left[\frac{1}{(e^{-a_1 u} + 1)(e^{a_2(u-L)} + 1)}\right]\bigg|_{u=u^*} = 0 \tag{18}$$

After taking the derivative, we can rewrite this equation as

$$\frac{e^{(a_1 u^* + a_2(L - u^*))}\left[a_1(e^{a_2(L-u^*)} + 1) - a_2(e^{a_1 u^*} + 1)\right]}{(e^{a_1 u^*} + 1)^2(e^{a_2(L-u^*)} + 1)^2} = 0 \tag{19}$$

For the left-hand side to be equal to zero, the term in the square brackets in the numerator must be equal to zero, because all other terms in the expression are always positive. We next define the term in the square brackets as a function of $u$,

$$h(u) = a_1(e^{a_2(L-u)} + 1) - a_2(e^{a_1 u} + 1). \tag{20}$$

The number of roots of $h(u)$ is equal to the number of local extrema of $f_{\text{dsig-base}}(t)$.

To determine the number of roots of $h(u)$, we first note the following limiting properties:

$$\lim_{u \to -\infty} h(u) = \infty \tag{21}$$

$$\lim_{u \to \infty} h(u) = -\infty \tag{22}$$

According to the mean value theorem, $h(u)$ must thus have at least one root. Next, we note that $h(u)$ is strictly decreasing:

$$\frac{d}{du}h(u) = -a_1 a_2 e^{a_2(L-u)} - a_1 a_2 e^{a_1 u} < 0 \tag{23}$$

Since the derivative of $h(u)$ is not equal to zero for any $u$ we can use Rolle's Theorem to conclude that the function cannot have more than one root. In conclusion, $h(u)$ has exactly one root, which implies that $f_{\text{dsig-base}}(u)$ has exactly one local extremum.

Finally, we demonstrate that the local extremum of $f_{\text{dsig-base}}(t)$ is a local maximum, by inspecting the second derivative of $g(u)$ at $u = u^*$.

$$
\begin{aligned}
\frac{d^2}{du^2} g(u)\big|_{u=u^*} = {} & \frac{e^{a_1 u - 2a_2(L-u)}}{(e^{a_1 u} + 1)^3 (e^{-a_2(L-u)} + 1)^3} \times \left[ \left( a_1(e^{a_2(L-u)} + 1) - a_2(e^{a_1 u} + 1) \right)^2 \right. \\
& \left. - a_1^2 e^{a_1 u}(e^{a_2(L-u)} + 1)^2 - a_2^2(e^{a_1 u} + 1)^2 e^{a_2(L-u)} \right]\big|_{u=u^*}
\end{aligned}
\tag{24}
$$

The first term in the square brackets is equal to $h(u)^2$ and thus is zero at $u = u^*$. The two remaining terms inside the square brackets are strictly negative, and the term outside the square brackets is strictly positive. Therefore, $g(u^*) < 0$, which proves that the local extremum is a local maximum.

### Funding
This work was supported by National Institutes of Health grant R01 AI120560. The funders had no role in study design, data collection and analysis, decision to publish, or preparation of the manuscript.

### Grant Disclosures
The following grant information was disclosed by the authors:
National Institutes of Health: R01 AI120560.

### Competing Interests
Claus O. Wilke is an Academic Editor for PeerJ.

### Author Contributions
- M. Umut Caglar conceived and designed the experiments, performed the experiments, analyzed the data, contributed reagents/materials/analysis tools, wrote the paper, prepared figures and/or tables, reviewed drafts of the paper.
- Ashley I. Teufel contributed reagents/materials/analysis tools, wrote the paper, prepared figures and/or tables, reviewed drafts of the paper.
- Claus O. Wilke conceived and designed the experiments, contributed reagents/materials/analysis tools, wrote the paper, reviewed drafts of the paper.

### Data Availability
Github: https://github.com/wilkelab/sicegar.

### Supplemental Information
Supplemental information for this article can be found online at http://dx.doi.org/10.7717/peerj.4251#supplemental-information.

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
