# Peer review of "Sicegar: R package for sigmoidal and double-sigmoidal curve fitting"

_PeerJ, doi:10.7717/peerj.4251_

## Round 0.1 · original submission · Major Revisions

Both of the reviewers have commented on several positive aspects of this manuscript as well as making what I see as constructive suggestions to further enhance its usefulness and rigor. In particular, I would like to highlight their suggestions around showing an application/example as I feel this would particularly enhance the manuscript's value. The brevity of the current version of the manuscript is one of its strengths but I think this material could be added without unduly compromising this.

Reviewer 1 ·

Basic reporting

See "General comments for the author"

Experimental design

See "General comments for the author"

Validity of the findings

See "General comments for the author"

Additional comments

The manuscript „Sicegar: R package for sigmoidal and double-sigmoidal curve fitting“ by Caglar and coworkers describes an R package that conducts sigmoidal curve fitting and model selection to choose between “sigmoidal”, “double sigmoidal”, “ambiguous” (no decision can be made) and “no signal” (no growth characteristics) curve structure. The current implementation is based on the following steps: i) normalizing the data within [0, 1], ii) checking for minimum and maximum threshold levels, iii) fitting a sigmoidal and double sigmoidal model by maximum likelihood (nonlinear least-squares) using the minpack.lm package and a grid of starting estimates for robustness, and finally iv) model selection based on lowest AIC. This algorithm is tested by an extensive simulation regime, in which increasing noise is added to perfect sigmoidal/double sigmoidal models (“ground truth”). It could be demonstrated that with the exception of extremely high noise levels, the implementation succeeds in identifying the underlying true model.
The manuscript is well-written and the package works well, however there are some important points the authors should address before rendering it suitable for publication.
1) It is not really clear - and nowhere mentioned - in which scenario (biological, pharmaceutical or clinical question) the selection between sigmoids/double sigmoids is essential. This information (maybe discriminating growth from growth with subsequent decay) should be supplied in the “Introduction”; else, the readership will see no real-world application.
2) AIC is not explained and not written in unabbreviated form.
3) When drawing the starting parameters from a uniform distribution (line 49), does it make sense to draw random values from it? Functions such as runif might miss some essential values. Would it not be better to create a sequence from within this window, i.e. seq(lower, upper, length.out = 100) to ensure that the complete parameter space is employed?
4) The authors should at least verify that the quality of model selection is also achieved when i) the predictor values are not an equidistant sequence (3 to 30, in 0.5 increments) and ii) the noise setup is heteroscedastic, i.e. noise is a function of the magnitude. This must not necessarily be included as a figure, but the outcome would be interesting as most of the real-world data does not have constant (homoscedastic) noise.
5) It would also be interesting to know how the model selection performs when the authors use a 5-parameter (Richards) sigmoidal model, which often fits better when the curve is asymmetric.
6) In lines 172ff, the authors describe a decreasing performance of model selection in a high-noise regime. In this context, the authors should cite the work from Spiess & Neumeyer (https://bmcpharma.biomedcentral.com/articles/10.1186/1471-2210-10-6), which investigated model sigmoidal model selection and resulting goodness-of-fit measures at varying noise levels.
7) I strongly believe that Fig. 1 and Fig. 2 could be merged into one concise Figure, i.e. the different phases of the double-sigmoid could be denoted in Fig. 2B.
8) The authors have completely neglected the drc package, which also offers a wealth of functionality for sigmoidal dose-response curve fitting and model selection. It should be cited.

·

Basic reporting

1) The introduction would need to be improved: it is simply too short (one (self) citation is not enough).

The authors need to provide background paragraphs on 1) existing (statistical) methods for fitting (nonlinear) s-shaped data (e.g., generalized linear and nonlinear regression) and 2) (more importantly) existing software for fitting s-shaped data. There are several R packages that should be mentioned: drc, drfit, grofit, qpcr, some of which are mentioned in the later subsection in lines 188-205; these lines should be integrated into the introduction. Note that the R package "drc", which can also fit biphasic patterns just like the proposed double sigmoidal model, is widely used for analysis for many different types of data.

2) There are several claims throughout the manuscript that would need to be substantiated by suitable references, e.g., lines 21-23, 25-26, 64 (Why these thresholds?), 132 (Fermi?), 159-160 (Any reference for this definition as it's an unusual definition. Usually such a range would be defined based on a range on the y axis, e.g., EC5 to EC95).

3) Definitions and methods are placed in the Results section. Please consider re-arranging parts of the material in the manuscript, e.g., lines 107-164 should be placed in the Methods section. On the other hand lines 124-131 are very relevant in the Discussion.

4) Please do not repeat definitions and methods used in the Results section, e.g., lines 182-183. Typo: "insure" should be "ensure".

Experimental design

The manuscript addresses the problem of fitting high-throughput s-shaped data by providing an R package that allows classification of such data into 4 categories (including two types of nonlinear patterns) depending on the signal in the data. This is an interesting but also challenging problem.

Moreover, it seems that a novel multiplicative double sigmoidal model has been proposed although there could be links to models for independent action of mixtures.

However, some claims should be omitted: In lines 123, 132 it's claimed that any sigmoidal curve is fully determined by 3 parameters. This is simply not correct in general. There exist both four and five-parameter models for s-shaped data. Please look into the literature on logistic or log-logistic models. Actually, the authors define a four-parameter model in the appendix! Confusing.

In lines 59-60: Perhaps I don't understand, but why should the AIC be larger than -10? Depending on the data AIC can take on arbitrary values I believe. A sensible rationale (a reference?) need to be supplied for such a non-standard rule.

It would also be helpful if the authors could specify the probability distribution that is assumed when fitting models using maximum likelihood (line 47). Is it least squares estimation?

Validity of the findings

The proposed R package performs as would be expected: the more noise the more difficult the classification. This is nicely shown in Figure 3.

Would it be possible to include a real application? It is always to good to see a methodology used in practice.

The authors need to provide a short conclusion summarizing their findings and, possibly, experiences using the R package. Also, if no real data example can be provided, it should be mentioned as a limitation (as then it would seen that the package has not been used much in practice; good to know for potential users).

Note also that high-throughput data may be very heterogeneous, being a mixture of data with various noise levels. It seems that this realistic setting has not been explored for the R package "sicegar".

---

## Round 0.2 · accepted · Accept

The reviewers and I appreciate the revisions you have made and I feel that the manuscript is now suitable for publication. I look forward to perhaps seeing some of the extensions for the package you discuss in the future.

Reviewer 1 ·

Basic reporting

No comment.

Experimental design

No comment.

Validity of the findings

No comment

Additional comments

The authors have adequately addressed all raised points and the manuscript is now suitable for publication. I especially liked the now included thorough analysis of the different noise regimes.
One small grammar issue: Line 280: Unlike sicegar, ...

·

Basic reporting

The manuscript now complies with rules of basic reporting.

Experimental design

More details and explanations have been included. The manuscript reads well now.

Validity of the findings

A data example has been included, providing added value to the manuscript. All results (real data and simulations) fit together.

Additional comments

The manuscript is much improved (a detail: make sure poliovirus is written the same way throughout the manuscript). No further comments.